# WEAK-TO-STRONG ENHANCED VISION MODEL

## ABSTRACT

Recent advancements in large language and vision models have demonstrated extraordinary capabilities, driving researchers to train increasingly larger models in pursuit of even greater performance. However, smaller, easier-to-train models often exist prior to these larger models. In this paper, we explore how to effectively leverage these smaller, weaker models to assist in training larger, stronger models. Specifically, we investigate the concept of weak-to-strong knowledge distillation within vision models, where a weaker model supervises a stronger one, aiming to enhance the latter's performance beyond the limitations of the former. To this end, we introduce a novel, adaptively adjustable loss function that dynamically calibrates the weaker model's supervision based on the discrepancy between soft labels and hard labels. This dynamic adjustment allows the weaker model to provide more effective guidance during training. Our comprehensive experiments span various scenarios, including few-shot learning, transfer learning, noisy label learning, and common knowledge distillation settings. The results are compelling: our approach not only surpasses benchmarks set by strong-to-strong distillation but also exceeds the performance of fine-tuning strong models on full datasets. These findings highlight the significant potential of weak-to-strong distillation, demonstrating its ability to substantially enhance vision model performance. Code will be released.

## 1 INTRODUCTION

> "Big things have small beginnings." — *Movie "Prometheus"*.

This adage aptly encapsulates the developmental journey of high-performance models in the fields of computer vision and natural language processing. The remarkable models that currently drive advancements in these areas did not appear out of nowhere; they evolved incrementally from simpler, less powerful architectures.

In the realm of NLP, the journey began with models like RNN and LSTM networks. These early models laid the foundation for more advanced architectures, gradually evolving into models like GPT (Radford et al., 2019; Brown et al., 2020). With its 175 billion parameters, GPT-3 showcased the transformative power of scaling up, ultimately paving the way for today's state-of-the-art large language models that excel in various tasks, from translation to creative writing. Similarly, the evolution in vision began with the pioneering LeNet (LeCun et al., 1998) architecture, designed for digit recognition. ResNet (He et al., 2016) then addressed the vanishing gradient problem, allowing for much deeper networks. Today, large vision models like ViTs (Dosovitskiy et al., 2020) continue to push the boundaries, achieving unprecedented performance surpassing human across various visual tasks.

As demonstrated by empirical studies on scaling laws (Kaplan et al., 2020), model performance typically scales with model size, dataset size, and the amount of compute used for training. This suggests that training larger models with more data holds the greatest potential for improvement, while other factors like training recipes or network architectures have relatively minimal impact across a wide range. However, before training a new large model, there often exists a smaller, weaker model. It's natural to ask whether these existing weaker models can be leveraged to assist in training larger ones. In this paper, we focus on addressing the challenge of how to efficiently utilize these weaker models to optimize and guide the training of more powerful, larger models.

Building on the notion of leveraging existing models, previous work has introduced the concept of "superalignment" to address the challenge of incorporating human expertise into the supervision of superhuman AI models. This approach seeks to align powerful models with human input to maximize their learning potential. A particularly relevant study in this context is Weak-to-Strong Generalization (Burns et al., 2023), which explores the intriguing possibility of using weaker models to supervise stronger ones. The findings are compelling: despite their inherent limitations, weaker models can provide supervision that enables stronger models — already equipped with superior generalization and representational power — to surpass their weaker counterparts. Remarkably, even when the weaker models offer incomplete or noisy labels, the stronger models are able to transcend these shortcomings, achieving higher performance. This concept has shown its efficacy in fields such as natural language processing and reinforcement learning, affirming the potential of Weak-to-Strong knowledge distillation as a viable and effective strategy.

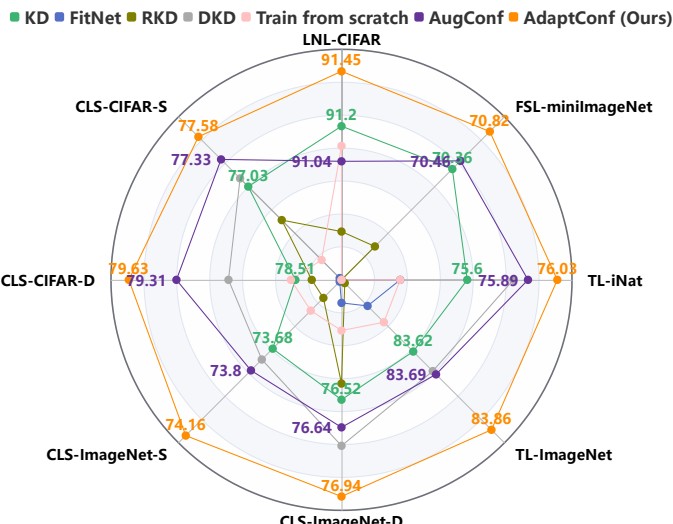

Figure 1: Our proposed AdaptConf achieves the best performance on a broad range of tasks compared with other knowledge distillation based methods. The corresponding values are calculated by averaging results on each task, *i.e.*, classification, transfer learning, few-shot learning, and learning with noisy labels. CLS-CIFAR-S (same model family): Table 2, CLS-CIFAR-D (different model family): Table 4a, CLS-ImageNet-S: Table 3, CLS-ImageNet-D: Table 3, TL-ImageNet: Table 7a, TL-iNat: Table 7b, FSL-miniImageNet: Table 5, LNL-CIFAR: Table 8.

| Model | CIFAR-100 validation set | | | | Model | ImageNet validation set | | | |
|---|---|---|---|---|---|---|---|---|---|
| | #Params | Top-1 (%) | Δ (%) | Win (%) | | #Params | Top-1 (%) | Δ (%) | Win (%) |
| MobileNet-V2 | 0.8M | 66.9 | - | - | ResNet-18 | 11.7M | 69.8 | - | - |
| ResNet-56 | 0.9M | 72.9 | -6.0 | 8.4 | ResNet-34 | 21.8M | 73.5 | -3.7 | 4.8 |
| ResNet-110 | 1.7M | 74.8 | -7.9 | 7.4 | ResNet-50 | 25.6M | 76.2 | -6.4 | 3.9 |
| VGG-13 | 9.5M | 75.3 | -8.4 | 6.3 | DeiT-S | 22M | 79.9 | -10.1 | 3.4 |
| ResNet32×4 | 7.4M | 79.9 | -13.0 | 4.9 | DeiT-B | 86M | 81.8 | -12.0 | 3.1 |
| ViT-B$_{\uparrow 224}$ | 86M | 89.0 | -22.1 | 2.5 | DeiT-B$_{\uparrow 384}$ | 86M | 83.0 | -13.2 | 2.8 |

Table 1: Comparison between models on CIFAR-100 and ImageNet. Δ represents the performance gap between the baseline MobileNet-V2 / ResNet-18 and other stronger models. "Win" indicates the ratio of samples correctly classified by MobileNet-V2 / ResNet-18 but incorrectly classified by the other stronger models.

We delve into the benefits brought via "vision superalignment", specifically investigating the applicability of Weak-to-Strong enhancement (W2S) within the context of vision tasks. Take the image classification task as an example, models typically develop from small to large, and the cost of training a small (weak) model is far less than that of training a large (strong) model. As shown in Table 1, larger models generally perform better. However, even when a model is 100 times larger and the top-1 accuracy is 22% higher, there are still many samples correctly identified by the weak model but misclassified by the strong model. This indicates that there are always opportunities to boost performance by leveraging the weak model. Therefore, when attempting to train a large, strong model, a natural question arises: *how can we leverage existing weak models to achieve further performance gains?* Our study meticulously designs and examines multiple scenarios in computer vision, including few-shot learning, transfer learning, noisy label learning, and traditional knowledge

distillation settings. In these scenarios, stronger models are trained to learn from weaker models. Through detailed validation and comparative experiments, we demonstrate the feasibility of W2S in the visual domain. Furthermore, we introduce an improved and adaptive confidence scheme to enhance the efficacy of W2S. Our work validates the concept of weak-to-strong boosting in computer vision, representing a significant advancement in understanding and optimizing the interaction between strong and weak models. This approach has the potential to pave the way for groundbreaking advancements in achieving human-level expertise and even superhuman artificial intelligence.

## 2 RELATED WORKS

The pursuit of enhancing the performance of deep neural networks in computer vision has led to the development of the teacher-student learning paradigm (Hinton et al., 2015; Roth et al., 2023). This approach typically involves a stronger model (teacher) improving the performance of a weaker model (student), with extensive research focusing on optimizing the capabilities of the weaker model. Various strategies have been proposed to achieve this. For instance, (Romero et al., 2014) suggests that in addition to the output logits, incorporating intermediate layer features for supervision can significantly boost the student's learning. (Park et al., 2019) posits that the relationships between samples can serve as valuable supervisory information.

In a further refinement of this approach, (Zhao et al., 2022) redefines classical knowledge distillation (KD) loss, segmenting it into target-class and non-target-class distillation to balance the transfer of these two types of information more effectively. (Heo et al., 2019) delves into the details and components of feature distillation, arriving at an improved method for the transfer of feature knowledge. Meanwhile, (Chen et al., 2021a) explores cross-stage feature transfer as an alternative to the conventional same-stage feature transfer. These methods have proven effective for strong-to-weak generalization scenarios.

However, with the gradual increase in the size and complexity of vision foundation models, the focus has shifted towards weak-to-strong boosting, *i.e.*, how a weak model can improve a strong model. In this context, (Furlanello et al., 2018) investigates knowledge distillation between teachers and students of equal size, demonstrating the feasibility of distilling models of the same size. Building upon this, (Xie et al., 2020) introduces the use of additional unlabeled data for knowledge distillation among models of equal size, further validating the effectiveness of strong-to-strong boosting, especially in scenarios with abundant data availability. This body of research sets the stage for our exploration into weak-to-strong boosting, a relatively uncharted yet promising domain in the field of vision foundation models.

Part of our experimental settings are similar to weakly supervised learning (Durand et al., 2017; Joulin et al., 2016), where there are no annotations (ground truth labels) in training machine learning models. However, unlike weak supervision, which focuses on obtaining a large amount of annotated data at a low cost, we are more interested in the weak-to-strong boosting process itself rather than the availability of annotations.

## 3 WEAK-TO-STRONG ENHANCEMENT

To advance towards super-human AGI models, a weak-to-strong approach is critical. This means using human-level intelligence as a foundation to guide and refine the development of more advanced, super-human systems. We begin by focusing on foundational tasks and models that can progressively support the growth of stronger architectures. In the following, we examine the feasibility of *weak-to-strong enhancement*, where simpler, weaker models provide useful supervision to more complex models. This step-by-step approach ensures that as models evolve, they can benefit from earlier stages, steadily improving in both capability and accuracy. One of the key challenges in this weak-to-strong framework is dealing with the noisy or incomplete supervision signals provided by weaker models. To mitigate this, we introduce a novel technique: *adaptive confidence distillation*. This method leverages the insights from weaker models while dynamically adjusting the level of trust placed in them. By modulating the influence of weak supervision based on its confidence, our approach ensures that the stronger model can benefit from imperfect labels without being misled. This adaptive mechanism allows stronger models to distill meaningful knowledge, even from less accurate outputs.

## 3.1 SELECTION OF VISION MODEL

In our exploration of weak-to-strong enhancement for vision models, it is essential to first define which models are suitable for this foundational research. Several categories of models can serve as strong candidates for vision foundation models, including text-vision models (Radford et al., 2021), image generation models (Rombach et al., 2022; Chen et al., 2020), and general zero-shot models (Bai et al., 2023; Kirillov et al., 2023). Each of these models brings unique strengths and approaches to solving computer vision tasks. To identify the most appropriate models for weak-to-strong boosting, we propose a definition that emphasizes both versatility and effectiveness. Vision foundation models should be capable of addressing a wide range of visual tasks while consistently delivering high-quality performance.

Based on these criteria, we propose that backbones pretrained on ImageNet are strong candidates for vision foundation models. The rationale behind this choice is threefold. First, ImageNet-pretrained backbones have consistently proven to be highly adaptable and effective across downstream vision tasks such as classification, detection, segmentation, tracking, colorization, *etc.*. Fine-tuning these backbones often results in state-of-the-art performance, underscoring their robustness and versatility. Second, there is a wealth of pretraining algorithms specifically developed for these models (He et al., 2022a; Xie et al., 2022), positioning them as universal tools for a variety of vision tasks. Furthermore, these models frequently serve as one branch in vision-language multimodal models (Du et al., 2022), further validating their applicability in cross-modal tasks. Finally, compared to other foundation models, such as CLIP (Radford et al., 2021) or Diffusion-based (Rombach et al., 2022) models trained on massive web-scale datasets, ImageNet-trained backbones are far more accessible. Their moderate computational demands make them a practical choice for a broader range of researchers with limited resources.

To validate the feasibility of weak-to-strong enhancement, we focus on these pretrained backbones and use image classification as the fundamental task. By selecting this well-established task, we create a controlled environment to rigorously test the effectiveness of our adaptive confidence distillation method. This will provide a solid baseline for expanding the weak-to-strong paradigm to more complex vision tasks and models in future work.

## 3.2 ADAPTIVE CONFIDENCE DISTILLATION

In this section, we explore the methodology for implementing weak-to-strong boosting in vision foundation models. The central question we address is how a weak vision foundation model can supervise a stronger counterpart effectively. (Burns et al., 2023) proposes an augmented confidence loss approach, which is formulated as:

$$L_{\text{conf}}(f) = (1 - \alpha)\text{CE}(f(x), f_w(x)) + \alpha\text{CE}(f(x), \hat{f}(x)), \quad (1)$$

where $f$ represent the strong model that needs to be optimized, and $f_w$ denote the weak model, $\hat{f}(x)$ refers to the hard label predicted by the strong model for an input image $x$. The loss function incorporates the cross-entropy loss (CE) and is balanced by a hyperparameter $\alpha$. In this formulation, the first term of the loss function resembles the traditional knowledge distillation loss, signifying the learning process of the strong model from the weak model. Given that the labels provided by the weak model may not always be accurate, the second term of the loss function encourages the strong model to leverage its superior generalization abilities and prior knowledge to refine its predictions.

The strength of this approach lies in its ability to balance direct learning from the weak model with the strong model's intrinsic capacity for understanding and interpreting the visual data. This method paves the way for the strong model to surpass the limitations of the weak model, utilizing the latter's guidance while simultaneously enhancing its predictions through its advanced capabilities.

Addressing the limitations inherent in the supervision provided by weak models and the inaccuracies of strong models' self-generated hard labels, a more sophisticated approach is required beyond a simple weighted combination of these labels. Given the challenge in directly discerning the accuracy of each label, leveraging confidence as a metric for selecting the most probable correct label emerges as a viable solution.

We propose to use the discrepancy between the soft label and the hard label as an indicator of the model's confidence. The underlying rationale is that when a model's soft label closely aligns with

| | ResNet20 | ResNet32 | ResNet8×4 | WRN-16-2 | WRN-40-1 | VGG8 |
|---|---|---|---|---|---|---|
| Teacher | 68.93 | 71.72 | 72.41 | 72.71 | 72.30 | 71.99 |
| | ResNet56 | ResNet110 | ResNet32×4 | WRN-40-2 | WRN-40-2 | VGG13 |
| Student | 72.94 | 74.80 | 79.90 | 77.20 | 77.20 | 75.26 |
| KD (Hinton et al., 2015) | 73.81 | 76.45 | 79.32 | 78.25 | 77.97 | 76.41 |
| FitNet (Romero et al., 2014) | 70.51 | 73.15 | 77.65 | 76.71 | 76.12 | 76.39 |
| RKD (Park et al., 2019) | 72.98 | 75.62 | 80.10 | 77.27 | 77.76 | 76.20 |
| ReviewKD (Chen et al., 2021a) | 70.15 | 72.30 | 77.22 | 75.86 | 75.78 | 74.22 |
| DKD (Zhao et al., 2022) | 73.90 | 76.57 | 79.52 | 78.18 | 77.95 | 76.62 |
| AugConf (Burns et al., 2023) | 73.86 | 76.72 | 80.34 | 78.34 | 78.15 | 76.55 |
| AdaptConf (**Ours**) | **74.17** | **76.86** | **80.64** | **78.58** | **78.40** | **76.84** |
| Δ | +1.23 | +2.06 | +0.74 | +1.38 | +1.20 | +1.58 |

Table 2: **Results on the CIFAR-100 validation set.** Teachers and students are in the **same** architectures. And Δ represents the performance improvement over the student model trained from scratch. All results are the average over 3 trials.

its hard label, it suggests a higher confidence in its own judgment. To capitalize on this insight, we introduce an adaptive confidence loss that dynamically adjusts based on the model's confidence level. The specific formulation of this loss is as follows:

$$L_{AC}(f) = (1 - \beta(x))\text{CE}(f(x), f_w(x)) + \beta(x)\text{CE}(f(x), \hat{f}(x)),$$

$$\beta(x) = \frac{\exp(\text{CE}(f(x), \hat{f}(x)))}{\exp(\text{CE}(f(x), \hat{f}(x))) + \exp(\text{CE}(f(x), \hat{f}_w(x)))}. \tag{2}$$

In this formula, $\beta(x)$ is a function of the input image $x$ that calculates the confidence weight and $\hat{f}_w(x)$ is the hard label of $x$ in the weak model. This weight determines the balance between learning from the weak model and relying on the strong model's own predictions. The cross-entropy loss (CE) is used for both components, with the first term focusing on learning from the weak model and the second term emphasizing the strong model's self-supervision.

This adaptive confidence loss enables a more nuanced approach to weak-to-strong boosting. By adjusting the weight based on confidence levels, it allows the strong model to discern when to prioritize its own predictions over the guidance of the weak model and vice versa. This adaptability is key to overcoming the inaccuracies and limitations of both models, leading to more effective learning and enhanced performance in vision foundation models.

## 4 EXPERIMENT

In this section, we report our main empirical results on various tasks, including baselines and promising methods. All implementation details are attached in supplementary materials.

### 4.1 TASKS

**Image Classification.** Our experiments are primarily focused on two benchmark datasets. CIFAR-100 (Krizhevsky et al., 2009) is a widely recognized dataset for image classification, comprising 32×32 pixel images across 100 categories, with training and validation sets containing 50,000 and 10,000 images, respectively. Conversely, ImageNet (Deng et al., 2009) is a large-scale dataset for classification tasks, encompassing 1.28 million training images and 50,000 validation images across 1,000 classes. Additionally, we explore scenarios where only soft labels generated by a weak teacher model are available for training.

| | ResNet18 | MobileNet-V1 |
|---|---|---|
| Teacher | 69.75 | 71.57 |
| | ResNet34 | ResNet50 |
| Student | 73.47 | 76.22 |
| KD (Hinton et al., 2015) | 73.68 | 76.52 |
| FitNet (Romero et al., 2014) | 70.93 | 73.61 |
| RKD (Park et al., 2019) | 73.65 | 76.45 |
| ReviewKD (Chen et al., 2021a) | 72.99 | 75.28 |
| DKD (Zhao et al., 2022) | 73.74 | 76.72 |
| AugConf (Burns et al., 2023) | 73.80 | 76.64 |
| AdaptConf (**Ours**) | **74.16** | **76.94** |
| Δ | +0.69 | +0.72 |

Table 3: **Top-1 results on the ImageNet validation set.** Δ represents the performance improvement over the student model trained from scratch.

| | ShuffleNet-V1 | ShuffleNet-V1 | MobileNet-V2 | MobileNet-V2 | ShuffleNet-V2 |
|---|---|---|---|---|---|
| Teacher | 72.40 | 72.40 | 66.85 | 66.85 | 74.44 |
| | ResNet32×4 | WRN-40-2 | VGG13 | ResNet50 | ResNet32×4 |
| Student | 79.90 | 77.20 | 75.26 | 80.43 | 79.90 |
| KD (Hinton et al., 2015) | 80.19 | 78.02 | 75.39 | 78.64 | 80.31 |
| FitNet (Romero et al., 2014) | 77.61 | 75.15 | 72.36 | 75.92 | 78.05 |
| RKD (Park et al., 2019) | 80.30 | 77.23 | 76.21 | 79.89 | 80.39 |
| ReviewKD (Chen et al., 2021a) | 78.43 | 75.98 | 73.69 | 77.05 | 77.84 |
| DKD (Zhao et al., 2022) | 80.55 | 78.10 | 75.81 | 79.65 | 80.67 |
| AugConf (Burns et al., 2023) | 80.62 | 77.92 | 76.43 | 80.75 | 80.84 |
| AdaptConf (**Ours**) | **80.99** | **78.55** | **76.58** | **80.98** | **81.06** |
| Δ | +1.09 | +1.35 | +1.32 | +0.55 | +1.16 |

(a) Trained with teacher's prediction and GT label. Δ is the improvement over the student trained from scratch.

| | ShuffleNet-V1 | ShuffleNet-V1 | MobileNet-V2 | MobileNet-V2 | ShuffleNet-V2 |
|---|---|---|---|---|---|
| Teacher | 72.40 | 72.40 | 66.85 | 66.85 | 74.44 |
| | ResNet32×4 | WRN-40-2 | VGG13 | ResNet50 | ResNet32×4 |
| Student | 79.90 | 77.20 | 75.26 | 80.43 | 79.90 |
| KD (Hinton et al., 2015) | 77.92 | 76.45 | 72.13 | 73.32 | 78.27 |
| FitNet (Romero et al., 2014) | 75.74 | 74.03 | 70.57 | 71.45 | 76.42 |
| RKD (Park et al., 2019) | 76.59 | 75.70 | 70.28 | 72.06 | 77.84 |
| AugConf (Burns et al., 2023) | 78.25 | 76.37 | 72.51 | 74.48 | 78.81 |
| AdaptConf (**Ours**) | **78.48** | **76.66** | **72.93** | **74.67** | **79.04** |
| Δ | +6.08 | +4.26 | +6.08 | +7.82 | +4.37 |

(b) Trained with teacher's prediction only. Δ represents the performance improvement over the teacher model.

Table 4: **Results on the CIFAR-100 validation set.** Teachers and students are in the **different** architectures. All results are the average over 3 trials.

**Few-shot learning.** We explore few-shot learning across the miniImageNet (Vinyals et al., 2016) dataset which contains 100 classes sampled from ILSVRC-2012 (Russakovsky et al., 2015). We randomly split the dataset into 64, 16, and 20 classes as training, validation, and testing sets, respectively. And ensure that each class has 600 images of 84×84 image size. We utilize the ResNet36 to explore the weak-to-strong boosting performance in few-shot task. To demonstrate weak-to-strong boosting performance, we follow Meta-Baseline and conduct related experiments on classifier stage and meta stage.

**Transfer learning.** We explore transfer learning across two benchmark datasets: ImageNet (Deng et al., 2009), and iNaturalist 2018 (Van Horn et al., 2018), the latter comprising 437,513 training images and 24,426 test images distributed across 8,142 species. We utilize the ViT-B (Dosovitskiy et al., 2020) that has been pretrained on the ImageNet training set using the self-supervised MAE (He et al., 2022b) approach, leveraging only image data without labels. Our results are reported for the fine-tuning phase, which is conducted under the guidance of a weak teacher model on each benchmark. Furthermore, we investigate scenarios where only soft labels produced by the weak teacher model are used for training.

**Learning with noisy labels.** We evaluate our approach using two datasets with simulated label noise, specifically CIFAR-10 (Krizhevsky et al., 2009) and CIFAR-100 (Krizhevsky et al., 2009). Consistent with prior research (Li et al., 2020; Tanaka et al., 2018), we introduce two distinct types of simulated noisy labels: symmetric and asymmetric. Symmetric noise is introduced by randomly substituting the labels of a certain proportion of the training data with other possible labels uniformly. In contrast, asymmetric noise involves systematic mislabeling to mimic real-world errors, such as flipping the labels to closely related classes. For example, in CIFAR-10, *truck* is mislabeled as *au-*

| | ResNet12 | | ResNet18 | |
|---|---|---|---|---|
| Teacher | 59.65 | 77.80 | 60.83 | 78.96 |
| | ResNet36 | | ResNet36 | |
| Student | 60.91 | 79.01 | 60.91 | 79.01 |
| | 1-shot | 5-shot | 1-shot | 5-shot |
| KD | 60.94 | 79.14 | 61.57 | 79.79 |
| RKD (Park et al., 2019) | 59.74 | 78.30 | 60.80 | 78.82 |
| AugConf (Burns et al., 2023) | 61.38 | 79.33 | 61.66 | 79.46 |
| AdaptConf (**Ours**) | **61.50** | **79.52** | **62.29** | **79.96** |
| Δ | +2.59 | +2.67 | +3.38 | +3.11 |

Table 5: **Average 5-way accuracy (%) with 95% confidence interval on the miniImageNet validation set in Classification Training stage.** Δ represents the performance improvement over the student model trained from scratch. All results are the average over 3 trials.

|  | Class-stage | | Meta-stage | |
|---|---|---|---|---|
| Teacher | ResNet12 | ResNet18 | ResNet12 | ResNet18 |
|  | 59.20 | 60.63 | 65.26 | 66.51 |
| Student | ResNet36 | ResNet36 | ResNet36 | ResNet36 |
|  | 65.08 | 65.08 | 65.08 | 65.08 |
| KD (Hinton et al., 2015) | 63.43 | 65.04 | 66.08 | 65.93 |
| RKD (Park et al., 2019) | 64.79 | 65.42 | 65.96 | 65.46 |
| AugConf (Burns et al., 2023) | 65.15 | 65.59 | 65.9 | 65.78 |
| AdaptConf (**Ours**) | **65.38** | **65.74** | textbf66.08 | **65.95** |
| Δ | +0.30 | +0.66 | +1.00 | +0.87 |

Table 6: **Average 5-way accuracy on miniImageNet validation set at Meta-Learning stage.** Δ represents the performance improvement over the student trained from scratch. All results are the average over 3 trials.

*tomobile*, *bird* as *airplane*, and *cat* is interchanged with *dog*. For CIFAR-100, similar mislabeling is applied within each of the super-classes in a circular fashion.

**Baseline methods.** The predominant framework for implementing teacher-student training paradigms is knowledge distillation (Hinton et al., 2015). This approach outlines a method where a larger, more complex teacher network guides the training of a more compact student network. Nonetheless, inspired by the findings of Burns *et al.* (Burns et al., 2023), our work pivots towards a scenario where the student network surpasses the teacher in visual capabilities. Despite this inversion of roles, there remains valuable dark knowledge in the teacher that can be transferred to the student, either through logits or via intermediate representational features. To benchmark our experiments, we employ a range of established (Hinton et al., 2015; Romero et al., 2014; Park et al., 2019; Heo et al., 2019; Chen et al., 2021a; Hao et al., 2023a) and recently proposed (Zhao et al., 2022; Burns et al., 2023) distillation techniques as baseline methods.

## 4.2 MAIN RESULTS

### 4.2.1 IMAGE CLASSIFICATION.

**CIFAR-100 image classification.** We commence our investigation with an exploration of weak-to-strong boosting (W2S) on the CIFAR-100 dataset. The outcomes of this investigation are delineated in Tables 2 and 4. Specifically, Table 2 presents the scenarios in which both teacher and student models share the same network architectures. We examine a range of prevalent vision architectures such as ResNet (He et al., 2016), WRN (Zagoruyko & Komodakis, 2016), and VGG (Simonyan & Zisserman, 2014). We employ various KD methods to assess the potential of larger-capacity students guided by limited-capacity teachers. Remarkably, in nearly all cases employing KD-based approaches, the student models outperform those trained from scratch.

Furthermore, both AugConf (Burns et al., 2023) and our proposed Adapt-Conf method surpasses all previous dis-

| Teacher: ResNet50 (80.36) | Teacher + GT | Teacher |
|---|---|---|
| Student: ViT-B (MAE pretrain) | 83.53 | - |
| KD (Hinton et al., 2015) | 83.62 | 82.32 |
| FitNet (Romero et al., 2014) | 82.48 | 81.02 |
| RKD (Park et al., 2019) | 82.19 | 80.98 |
| DKD (Zhao et al., 2022) | 83.68 | - |
| AugConf (Burns et al., 2023) | 83.70 | 82.38 |
| AdaptConf (**Ours**) | **83.86** | **82.51** |
| Δ | +0.33 | +2.15 |

(a) Top-1 results on the ImageNet validation set.

| Teacher: ResNet101 (67.42) | Teacher + GT | Teacher |
|---|---|---|
| Student: ViT-B (MAE pretrain) | 75.28 | - |
| KD (Hinton et al., 2015) | 75.60 | 71.57 |
| FitNet (Romero et al., 2014) | 73.68 | 70.11 |
| DKD (Zhao et al., 2022) | 75.82 | - |
| AugConf (Burns et al., 2023) | 75.90 | 71.73 |
| AdaptConf (**Ours**) | **76.03** | **71.99** |
| Δ | +0.75 | +4.57 |

(b) Top-1 results on the iNaturalist 2019 test set.

Table 7: **Transfer learning results.** The student model is a ViT-B (Dosovitskiy et al., 2020) pretrained by the self-supervised MAE framework (He et al., 2022b). Δ denotes the performance improvement over student/teacher in second/third columns.

tillation techniques across all teacher-student pairs. This highlights that simply emulating a weak teacher does not yield the most favorable outcomes. Notably, AdaptConf consistently achieves superior performance compared to AugConf (Burns et al., 2023), underscoring the advantage of our dynamic adaptive confidence weighting. This approach provides a more refined mechanism for facilitating weak-to-strong knowledge transfer.

| dataset | CIFAR-10 | | | | CIFAR-100 | | | |
|---|---|---|---|---|---|---|---|---|
| noise type | asymmetric | | symmetric | | asymmetric | | symmetric | |
| Teacher | PR18 | | PR18 | | PR18 | | PR18 | |
| | 92.98 | 99.56 | 95.80 | 99.80 | 73.20 | 92.67 | 76.16 | 92.90 |
| Student | PR34 | | PR34 | | PR34 | | PR34 | |
| | 93.69 | 99.61 | 96.13 | 99.77 | 74.80 | 92.94 | 78.20 | 93.77 |
| | Top-1 | Top-5 | Top-1 | Top-5 | Top-1 | Top-5 | Top-1 | Top-5 |
| KD (Hinton et al., 2015) | 93.54 | 99.84 | 95.90 | 99.84 | 75.49 | 93.67 | 77.61 | 93.74 |
| RKD (Park et al., 2019) | 92.42 | 99.75 | 95.99 | 99.85 | 74.20 | 93.54 | 76.92 | 93.09 |
| AugConf (Burns et al., 2023) | 92.60 | 99.75 | 95.10 | 99.83 | 74.99 | 93.72 | 78.34 | 94.02 |
| AdaptConf (**Ours**) | **93.69** | **99.84** | textbf96.13 | **99.87** | **75.61** | **93.78** | **78.64** | **94.03** |
| $\Delta$ | +0.00 | +0.23 | +0.00 | +0.10 | +0.81 | +0.84 | +0.44 | +0.26 |

Table 8: **Top-1 and top-5 results on the CIFAR-10/CIFAR-100 noise label validation set.** $\Delta$ represents the performance improvement over the student model trained from scratch. All results are the average over 3 trials.

Table 4 shows the results of teacher-student pairs from different series, such as ShuffleNet (Zhang et al., 2018) and MobileNet (Sandler et al., 2018). Additionally, take the MobileNetV2-ResNet50 pair as an example, the experimental results reveal that when the teacher model is significantly weaker, *i.e.*, a substantial performance gap exists between the weak teacher model and the strong student model, none of the KD-based methods were able to effectively enhance the strong student's performance, except for AugConf and AdaptConf. The possible reason is that these methods include the predictions of the strong student in the loss function. This proves that self-training methods, akin to those described in (Lee et al., 2013), can mitigate the bias from a suboptimal teacher model. It is important to note that FitNet (Romero et al., 2014) consistently underperforms when compared to training from scratch. This could be attributed to its sole focus on intermediate features, which may be more misleading for the strong student to learn from than soft predictions, as suggested by (Hao et al., 2023b). Overall, our AdaptConf achieves an improvement of 0.5%-2% on all evaluated teacher-student pairings, whether they are from the same or different series.

Furthermore, we investigate a scenario where only the teacher's output is available, as shown in Table 4b. In this context, it becomes evident that AugConf and AdaptConf yields more significant improvements compared to other KD-based methods when ground truth is absent. This observation underscores the suitability of our confidence distillation approach for more extreme W2S scenarios where ground truth is not available.

**ImageNet image classification.** Table 3 presents the top-1 accuracy for image classification on the ImageNet dataset. Our AdaptConf method achieves significant improvements across both W2S scenarios, whether employing the same or different architectures.

### 4.2.2 FEW-SHOT LEARNING

For the few-shot learning task, we conduct distillation experiments separately in the classification (Table 5) and meta-learning (Table 6) stages. We compare and evaluate the performances of student when trained with teachers of different sizes. In the classification experiments, only RKD results in a performance degradation of the student model, while the usage of other methods led to varying degrees of improvement. Notably, our confidence-based method outperforms previous knowledge distillation based ones. In the meta-learning stage, we employ weights from different training stages of the same model as the teacher. Experimental results demonstrate significant advantages of our proposed method. Even when using the Class-stage weight as the teacher, our approach achieves a +0.66% improvement over the baseline set by a weaker ResNet18 (Class-stage) teacher model. Furthermore, when using the same stage weight as the teacher, our confidence-based method surpasses previous knowledge distillation results to a greater extent.

### 4.2.3 TRANSFER LEARNING

Table 7 examines the efficacy of transfer learning using the iNaturalist (Van Horn et al., 2018) and ImageNet (Deng et al., 2009) datasets. When our method is trained with ground truth labels on ImageNet, it demonstrates a notable enhancement, achieving an increase of +0.33% in top-1 accuracy on a model with a high precision of 83.5%. Even without ground truth labels, our approach still secures a +2.15% improvement over the baseline set by a weaker ResNet50 teacher model. On

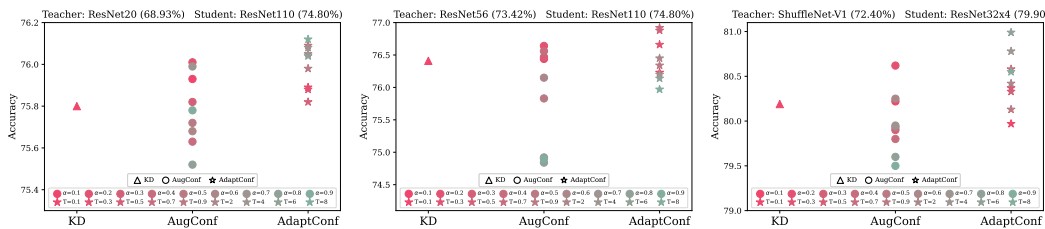

Figure 2: Ablation study examining the impact of hyper-parameter variation on confidence distillation results. The parameter $\alpha$ for AugConf is adjusted across a range from 0.1 to 0.9, while the temperature $T$ for AdaptConf is scaled from 0.1 to 8.

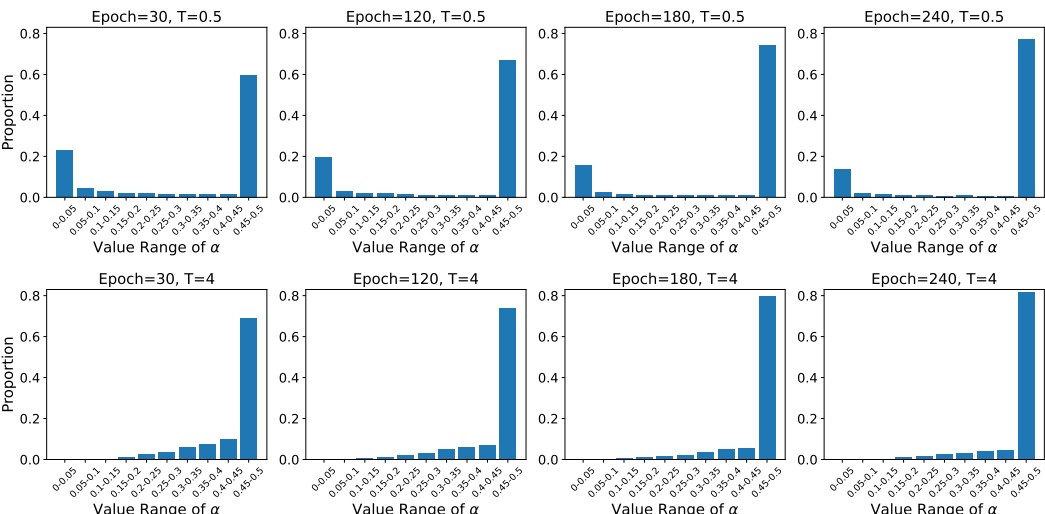

Figure 3: Quantitative analysis about the value of $\beta(x)$ in Eq. 2 on the CIFAR-100 dataset. The evaluation is based on the ShuffleNetV1-ResNet32x4 teacher-student architecture pair.

the iNaturalist dataset, our confidence-based method also surpasses previous knowledge distillation results by a considerable margin.

### 4.2.4 LEARNING WITH NOISY LABELS

In Table 8, we analyze the effectiveness of weak-to-strong using the CIFAR-10 and CIFAR-100 datasets under two simulated noisy label settings. When training the model on the sample dataset (CIFAR-10), all methods except ours, negatively impact the model given its already high accuracy. This underscores the robustness of our method, irrespective of the performance gap between the teacher and student models. On the CIFAR-100 dataset, our method demonstrates a performance improvement of 0.81% in top-1 accuracy under the asymmetric noise type setting.

### 4.3 ABLATION STUDY

**Robustness of confidence distillation.** In this study, we investigate the necessity of devising a method that goes beyond a mere weighted combination of labels. As depicted in Eq. 1, despite its straightforward approach of integrating direct learning from a weaker model with the intrinsic capacity of a stronger model, AugConf (Burns et al., 2023) still requires manual tuning of a hyper-parameter $\alpha$ to balance the ratio of two different objectives. The setting of different $\alpha$ values can have varying impacts across different contexts. Similarly, although our proposed AdaptConf does not require manual adjustment of $\alpha$ to balance the proportions of objectives, we can manipulate the temperature $T$ to control the degree of probability distribution in soft labels during the computation of the cross-entropy $CE(\cdot)$, following a conventional distillation method (Hinton et al., 2015). Therefore, we explore the effects of these two methods under different hyper-parameter settings on

the final outcome. Overall, the performance of KD, AugConf, and AdaptConf improves sequentially across various architectural settings. Moreover, it can be observed that AugConf exhibits a larger fluctuation in results compared to AdaptConf, indicating that the influence of $\alpha$ on AugConf is more significant than the effect of $T$ on AdaptConf, which suggests that our AdaptConf has superior robustness. Additionally, the average outcomes achieved by AdaptConf are consistently higher than those of AugConf under different hyper-parameter settings.

**Robustness of confidence distillation.** In this section, we perform a quantitative analysis of the confidence weight determined by our dynamic function $\beta(x)$ as delineated in Eq. 2, with the findings illustrated in Figure 3. We selected checkpoints from four distinct training phases and calculated their specific $\beta(x)$ values on the validation set. It can be observed that as training progresses, the proportion of samples with $\beta = 0.5$ increases, indicating that the student model's performance is improving and being aligned with the weak teacher's correct classifications. A higher temperature setting $T$ reduces the cross-entropy (CE) discrepancy between the teacher and student, promoting a more uniform balance between the weak teacher's guidance and the strong student's own predictions. Consequently, the number of samples where $\beta = 0.5$ also increases with training. These phenomena collectively validate that our proposed AdaptConf can dynamically adjust the learning ratio between the two components.

## 5 CONCLUSION

In this paper, we investigate weak-to-strong boosting for vision foundation models and unveil a promising avenue for enhancing the capabilities of artificial intelligence in the visual domain. By leveraging an innovative adaptive confidence loss mechanism, we demonstrate the feasibility and effectiveness of using weaker models to supervise and improve stronger counterparts. Our findings not only validate the potential of weak-to-strong enhancement but also set the stage for future research endeavors aimed at unlocking further advancements in AI model performance. This work contributes a significant step forward in the pursuit of more sophisticated, efficient, and capable AI systems, emphasizing the importance of nuanced supervision mechanisms in achieving better performance in vision tasks.

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

## A APPENDIX: IMPLEMENTATION DETAILS.

### A.1 IMAGENET CLASSIFICATION

**CIFAR-100.** We adopt the vision architectures of the teacher and student models as outlined in the traditional distillation papers (Hao et al., 2023b; Zhao et al., 2022). It should be noted that the previous codebase (Zhao et al., 2022) conducted experiments on CIFAR-100 using only 1 GPU. To expedite our experiments, we leverage the distributed Pytorch framework (Paszke et al., 2019) to train and do inference on 8 GPUs. Consequently, some hyperparameter settings and results may not

align exactly with the previous paper. Specifically, we employ the SGD optimizer with a momentum of 0.9. The learning rate starts at 0.2 and decays with a minimum learning rate of 2e-3 using a cosine annealing schedule. We train for 240 epochs with a batch size of 512 spread across 8 GPUs, and apply a weight decay of 0.0005. Standard data augmentation techniques, including random resized crop and horizontal flip are utilized.

**ImageNet.** we employ the SGD optimizer with a momentum of 0.9. The learning rate starts at 0.1 and decays with a rate of 0.1 every 30 epochs. We train for 100 epochs with a batch size of 512 spread across 8 GPUs, and apply a weight decay of 0.0001. Standard data augmentation techniques, including random resized crop, horizontal flip and label smoothing are utilized.

## A.2 Transfer learning.

To fine-tune the self-supervised pretrained ViT-B on ImageNet and iNaturalist, we adopt the hyperparameter settings from MAE (He et al., 2022b). The adamw optimizer is employed for this purpose. The learning rate begins at 2e-3 and gradually decays with a minimum learning rate of 1e-6, utilizing a cosine annealing schedule. We conduct training for 100 epochs, utilizing a batch size of 4096 across 8 GPUs. A weight decay of 0.05 is applied to mitigate overfitting. The fine-tuning process incorporates robust data augmentation techniques, including auto-augment, mixup, cutmix, and stochastic drop path.

## A.3 Few-shot Leaerning

We use ResNet12 and follow the setting of (Chen et al., 2021b) on miniImageNet dataset, and created ResNet18 and ResNet36 by increasing the number of layers in original ResNet12. For the classification training stage, we use the SGD optimizer with momentum 0.9. The learning rate starts from 0.1 and the decay factor is set to 0.1. On miniImageNet, we train 100 epochs with the batch size of 128 on 4 GPUs, the learning rate decays at 90 epoch, and the weight decay is 0.0005. Standard data augmentation strategies including random resized crop and horizontal flip are applied. For meta-learning stage, we use the SGD optimizer with momentum 0.9. The learning rate is fixed as 0.001. The batch size is set to 4, *i.e.*, each training batch contains 4 few-shot tasks to compute the average loss. The cosine scaling parameter $\tau$ is initialized as 10. For knowledge distillation, the kd loss weight is set to 1, the temperature is set to 10. We use the threshold with 8 and 0.25 for classifier stage and meta stage, respectively.

## A.4 Learning with noisy labels

For CIFAR-10/100 datasets, we follow (Li et al., 2022) use a PreAct ResNet18 network, and created PreAct ResNet34 by increasing the number of layers in PreAct ResNet12. We train our models using SGD with a momentum of 0.9, a weight decay of 1e4, and a batch size of 128. The network is trained for 250 epochs and the warm-up epoch is set to 1 dufring training stage. We set the initial learning rate as 0.1, and reduce it by a factor of 10 after 125 and 200 epochs. The fine-tuning stage of Sel-CL+ has 70 epochs, where the learning rate is 0.001. We always set the Mixup hyperparameter to 1, scalar temperature to 0.1, and loss weights to 1. We try two settings of simulated noisy labels: symmetric and asymmetric. And the noise ratio is set to 0.2 and 0.4, respectively. For knowledge distillation, we set the threshold to 0.5 and assign a weight of 1 to the knowledge distillation loss.

