# OpenReview forum: "Weak-to-Strong Enhanced Vision Model"
_ICLR.cc/2025/Conference — ICLR 2025 Conference Withdrawn Submission_

### Official Review · Reviewer_h6y4 · 2024-10-31

**Soundness:** 3
**Presentation:** 3
**Contribution:** 2
**Rating:** 5
**Confidence:** 4

**Summary:**

The authors propose that weak-to-strong distillation can aid in training a more powerful visual model. They introduce a weighting mechanism based on the difference between soft and hard labels to adjust the balance in distillation. Comprehensive experiments on various setups for image classification tasks validate that weak-to-strong distillation can enhance the performance of stronger model.

**Strengths:**

1. Through comprehensive experiments, this paper demonstrates that weak-to-strong distillation can raise the performance ceiling of strong models, offering a promising direction for research on knowledge distillation in the field of computer vision.

2. An adaptive mechanism is introduced to mitigate noise from weaker models and reduce the baseline’s hyperparameters, and the detailed comparisons are conducted against the baseline under various parameter settings.

3. The paper is well-written and easy to follow.

**Weaknesses:**

1. The technical novelty of this work is limited, as the weak-to-strong concept is derived from the baseline method [1], and the loss function in Eq. 1 was also proposed there. While this work offers an incremental improvement over the baseline, the tables do not clearly show the performance gains relative to it; including such data would improve clarity. Additionally, the proposed adaptive threshold mechanism delivers only modest performance gains (e.g., 0.25 on CLS-CIFAR-S and 0.17 on TL-ImageNet), and its motivation is not particularly convincing. A more thorough analysis is recommended.

2. It would be better if there is an overall comparative analysis of its benefits given fixed teacher model or student model. Specifically, the authors could refer to Figure 3 in [1] and Figure 2 in [2]. It is recommended to include a curve plot to demonstrate the trend of accuracy improvement for the student model. For one scenario, a strong student model should be fixed, with the y-axis representing its accuracy, the x-axis showing the parameter count of the teacher model, and different hues depicting various methods. Similarly, another plot could be provided by fixing the teacher model while varying the student model's parameters.

3. Table 1 indicates that the weak model correctly predicts some samples that the strong model misclassifies, suggesting that the weak model may improve the strong model. However, the experimental results do not show if the strong model's prediction accuracy for these samples improved after distillation. It is recommended that the authors analyze the accuracy variation of the student model for this subset of samples and provide examples of misclassified samples that were corrected.

Refs:

[1] Weak-to-strong generalization: Eliciting strong capabilities with weak supervision.

[2] Self-Training With Noisy Student Improves ImageNet Classification.

**Questions:**

In the knowledge distillation methods for comparison, the performance of a weak-to-strong model may not always match or surpass that of a strong student model. Gaining insight into the specific conditions under which knowledge distillation (KD) methods improve performance relative to the strong student model is crucial. Additionally, it's equally important to identify scenarios where KD methods do not enhance performance.
The authors could provide further analysis to clarify these scenarios.

---

### Official Review · Reviewer_ncvz · 2024-11-02

**Soundness:** 2
**Presentation:** 2
**Contribution:** 1
**Rating:** 3
**Confidence:** 5

**Summary:**

This paper presents an approach for training vision models by leveraging a weak-to-strong knowledge distillation framework, where a smaller model supervises a larger, more complex model. The authors introduce an adaptive confidence loss that aims to balance the strong model’s learning from the weak model’s outputs while dynamically adjusting this influence based on confidence levels. The proposed method is evaluated across tasks including few-shot learning, transfer learning, and noisy label scenarios, where it reportedly outperforms several established knowledge distillation baselines.

**Strengths:**

- The idea of reversing the traditional strong-to-weak knowledge distillation paradigm is intriguing and could inspire further work in this area.
- The adaptive confidence loss function introduces an interesting approach to dynamically modulate the influence of the weak model, addressing a common limitation of distillation from lower-capacity models.
- The paper reports results across multiple vision tasks and benchmarks, showing that the proposed method can outperform some existing techniques in specific scenarios.

**Weaknesses:**

- The paper lacks a solid justification for why weak-to-strong knowledge distillation should be effective, especially in scenarios where the weak model has substantial inaccuracies. Without this foundation, the adaptive confidence loss feels somewhat ad hoc, and the intuition behind its efficacy is not well substantiated.

- The paper’s motivation for using weak-to-strong distillation instead of strong-to-strong or standard knowledge distillation methods is unclear. The introduction provides limited insight into specific cases where a weak-to-strong approach would provide distinct advantages over using a stronger teacher, particularly given that weaker models inherently have reduced representational power.
Further, the authors do not address scenarios in which this approach might fail, which leaves concerns about its broader applicability.

- Although the experiments cover several tasks, they do not provide sufficient insights into how weak-to-strong distillation behaves across different data complexities or model architectures. It is unclear whether the adaptive confidence method generalizes well or if its performance gains are task-specific.

- Additionally, the comparative baselines, while present, lack depth in terms of strong-to-strong distillation baselines. The addition of strong baselines would be necessary to make a compelling case for weak-to-strong distillation.

- Why does the method limit its experiments to the vision-only domain? The approach sounds general enough for any domain.

- The paper lacks details in the implementation and hyperparameter choices, particularly in how the adaptive confidence mechanism is tuned across different tasks. This omission raises concerns about reproducibility, as it is unclear how much of the observed performance gain is due to specific tuning choices rather than the generalizability of the method.

- Overall, the method has limited novelty in terms of the methodology.

**Questions:**

See the weakness section.

---

### Official Review · Reviewer_SbzE · 2024-11-02

**Soundness:** 2
**Presentation:** 3
**Contribution:** 2
**Rating:** 3
**Confidence:** 3

**Summary:**

The authors introduce the idea of weak-to-strong knowledge distillation into vision models, i.e., using weak models to guide and enhance the performance of strong models. The authors first illustrate through experiments (Table1) that weak models still have aspects that are worth learning from strong models, then propose a new adaptive loss function to balance the relationship between weak and strong models, so that the strong model can learn from the weak model while reducing the misguidance of the weak model's error signals to the strong model, and finally validate the feasibility of their approach through experiments on multiple tasks.

**Strengths:**

1. decent paper formatting.

2. the method is simple yet effective.

**Weaknesses:**

1. Compared with AugConf, the author's proposed AdaptConf improvement is not that significant, only a small performance improvement.

2. Is it the first to introduce weak-to-strong knowledge distillation in visual models?  Clarification on this point would be appreciated. I see this method as a straightforward modification of AugConf. While I understand the intuitive design of β(x), I don't see any rigorous theoretical or experimental analysis supporting this choice, thus the novelty is somewhat limited to me.

3.  In Table 1, it is evident that as the model size increases, the win values are decreasing, indicating that the guidance provided by the smaller model to the larger model is diminishing. Could you elaborate on whether you anticipate that weak-to-strong knowledge distillation may become ineffective if the strong model continues to scale? Additionally, could there be a risk that it might negatively impact the performance of the strong model?

4. Besides, I'm concerned about how the method addresses the overconfidence issue in strong models, which I believe deserves a more thorough investigation.

5. From the paper and the results, I did not clear see the necessary for the weak-to-strong paradigm, as the models compared in the experiment section is not very big, and in fact it is easy to find bigger and stronger models. The necessity of adopting small models as the teachers is still not convincing to me at least.

6. The experimental results may not tell the superiority of the method. There is limited comparison with the method in the table as there are many methods published in 2024 and they should be compared and discussed, including but not limited to:

[1] VkD: Improving Knowledge Distillation using Orthogonal Projections. CVPR 2024
[2] Logit Standardization in Knowledge Distillation . CVPR 2024
[3] C2KD: Bridging the Modality Gap for Cross-Modal Knowledge Distillation. CVPR 2024

**Questions:**

Please address the concerns raised in the weakness section.

---

### Official Review · Reviewer_cwSQ · 2024-11-03

**Soundness:** 2
**Presentation:** 3
**Contribution:** 2
**Rating:** 6
**Confidence:** 4

**Summary:**

The submission explores how to leverage a weaker model (with a smaller model capacity) trained on a fixed dataset to improve a model with a bigger capacity. To achieve said goal, the authors propose to add soft labels from the weak model as additional supervision for training; to avoid learning from the mistakes of the weaker model, a confidence-based adaptive weighting scheme is proposed to weigh the different sources of supervision differently (i.e., when the student is not confident about its prediction, focus more on learning from the weaker teacher). Extensive experiments are conducted on myriads of image classification datasets (e.g., ImageNet, CIFAR) and problem settings (supervised, few-shot, transfer learning) to show the approach's efficacy.

**Strengths:**

[S1] Extensive experiment: The authors have done extensive experiments to show the approach's effectiveness

[S2] Simple and interesting finding: The proposed approach is simple, intuitive, and effective.

**Weaknesses:**

[W1] The key premise of the approach is to leverage the diversity of model predictions to achieve better performance. Another line of work with a similar idea is model ensembling, which shows that averaging predictions from multiple models would often yield superior results. The idea of supervision from a model with a smaller capacity would be extremely enticing if the smaller models were specialist models trained to perform well on a single task and the larger students were generalist models trained to perform well on many tasks. However, since both the teachers and students are trained to perform well on the same task, it is unclear to the reviewer whether supervision from a weak teacher is a better approach to leveraging the diversity of model predictions or if better performance can be attained by leveraging ensembling approaches such as [1, 2].



[1] Huang, Gao, Yixuan Li, Geoff Pleiss, Zhuang Liu, John E. Hopcroft, and Kilian Q. Weinberger. "Snapshot ensembles: Train 1, get m for free." arXiv preprint arXiv:1704.00109 (2017).

[2] Izmailov, Pavel, Dmitrii Podoprikhin, Timur Garipov, Dmitry Vetrov, and Andrew Gordon Wilson. "Averaging weights leads to wider optima and better generalization." arXiv preprint arXiv:1803.05407 (2018).

**Questions:**

Suggestions:

[S1] The reviewer would like to caution the authors against using superficial terms/sentences such as superhuman artificial intelligence (line 113) and super-human AGI (line 150). They make the paper sound fictional instead of technical.

[S2] The citation for metabaseline [line 299] is missing.

[S3] Section 3.1: The justifications for ImageNet seem unnecessary to the reviewer. Besides, they are not convincing as well. First, ImageNet pre-trained models have been around much longer than CLIP, so they are more widely adopted, but it is unclear whether they are better than CLIP. Second, CLIP is more commonly used in VLM research than ImageNet (see LLaVA variants). Third, CLIP is no less accessible than ImageNet since it can be accessible through common libraries such as Huggingface.



Question:

[Q1] Eqn 2 does not involve using GT at all. The reviewer assumes that there is some burn-in period in model training that the ground truth is used before switching to eqn 2. It would be great if the authors could clarify how the ground truth is used concerning eqn 2.

---

### Note · Authors · 2024-11-13

I have read and agree with the venue's withdrawal policy on behalf of myself and my co-authors.